# Suspended Multifunctional Nanocellulose as Additive for Mortars

**DOI:** 10.3390/nano12071093

**Published:** 2022-03-26

**Authors:** Maria Vittoria Diamanti, Cristina Tedeschi, Mariagiovanna Taccia, Giangiacomo Torri, Nicolò Massironi, Chiara Tognoli, Elena Vismara

**Affiliations:** 1Department of Chemistry, Materials and Chemical Engineering “Giulio Natta”, Politecnico di Milano, 20131 Milan, Italy; nicolo.massironi@polimi.it (N.M.); chiara.tognoli@polimi.it (C.T.); 2Department of Civil and Environmental Engineering, Politecnico di Milano, 20131 Milan, Italy; cristina.tedeschi@polimi.it (C.T.); mariagiovanna.taccia@polimi.it (M.T.); 3Istituto Scientifico di Chimica e Biochimica “Giuliana Ronzoni”, 20131 Milan, Italy; torri@ronzoni.it

**Keywords:** oxidized nanocellulose, TEMPO, acrylates, suspension, mortar, mechanical resistance, water absorption

## Abstract

Cellulose derivatives have found significant applications in composite materials, mainly because of the increased mechanical performance they ensure. When added to cement-based materials, either in the form of nanocrystals, nanofibrils or micro/nanofibers, cellulose acts on the mixture with fresh and hardened properties, affecting rheology, shrinkage, hydration, and the resulting mechanical properties, microstructure, and durability. Commercial cotton wool was selected as starting material to produce multifunctional nanocelluloses to test as additives for mortars. Cotton wool was oxidized to oxidized nanocellulose (ONC), a charged nanocellulose capable of electrostatic interaction, merging cellulose and nanoparticles properties. Oxidized nanocellulose (ONC) was further functionalized by a radical-based mechanism with glycidyl methacrylate (GMA) and with a mixture of GMA and the crosslinking agent ethylene glycol dimethacrylate (EGDMA) affording ONC-GMA and ONC-GMA-EGDMA, both multifunctional-charged nanocellulose merging cellulose and bound acrylates properties. In this work, only ONC was found to be properly suitable for suspension and addition to a commercial mortar to assess the variation in mechanical properties and water-mortar interactions as a consequence of the modified microstructure obtained. The addition of oxidized nanocellulose caused an alteration of mortar porosity, with a decreased percentage of porosity and pore size distribution shifted towards smaller pores, with a consequent increase in compressive resistance, decrease in water absorption coefficient, and increased percentage of micropores present in the material, indicating a potential improvement in mortar durability.

## 1. Introduction

Polysaccharides are probably the most flexible and incredible biopolymer class due to the enormous numbers of properties they possess. Among them, two significant examples are cellulose, which is an important structural component of the primary cell wall of green plants and heparin, which is a life-saving drug. It is not crazy to match cellulose with heparin. Although both cellulose and heparin are roughly speaking made by a glucose unit skeleton, the roles they can play are different depending on the skeleton variation. Why are they so different? First, cellulose is neutral and heparin is highly negatively charged. More generally, we can argue that electrostatic forces and consequently electrostatic interactions divert the behavior of whatever compounds toward their peculiar material property.

The chemistry of cellulose, similarly to heparin, is the glucose reactivity. The glucose ring creates many different structures by involving the OH group reactivity and/or by reacting on C-H (OH) group. Cellulose reactivity regards these glucose groups. As any glucose unit carries three OH groups and seven C-H groups, it is difficult to think of selective reactions on cellulose. The second aspect of cellulose reactivity concerns solubility, or better put, its insolubility. Finally, morphology of cellulose is very complex and dramatically changes depending on the cellulose natural source. A paradigm of cellulose reactivity is acetylation where total esterification transforms the insoluble crystalline cellulose into the amorphous soluble cellulose acetate of multiple applications.

Cellulose is the precursor of a groundbreaking nanomaterial, the nanocellulose. Scientists envisage nanocellulose (NC) as one of the most promising green materials due to its intrinsic properties, renewability, and abundance [1,2]. The association of a high surface area with mechanical properties such as high modulus and tensile strength justifies the interest in many NC applications, not only as it is, but also as a nanocomponent of composite materials [3,4,5,6].

More recently, NC used as additives for cement-based materials were explored. NC has been studied as a mortar additive due to its great environmental appeal because it is a biodegradable and non-petroleum-based material. Renewability and sustainability aspects are clearly extremely attractive in a field that strongly suffers from an excess of environmental impact, such as the cement industry.

The first literature works in the field were mostly about the addition of microcellulose [7,8,9]; the exclusive use of NC is more recent [10,11,12], as a result of extensive efforts in reducing clogging due to flocculation issues. Indeed, NC demonstrated potential in modifying both fresh and hardened properties of concrete and mortars, from the reduction of bleeding and segregation phenomena [13], to enhanced cement paste curing and, more generally speaking, strengthening effects [14,15] and to self-healing [16]. Moreover, Guo et al. summarized the effects of cellulose nanocrystals (CNC), cellulose nanofibrils (CNF), bacterial cellulose (BC), and cellulose filaments (CF) not only on the mechanical resistance of cementitious materials, but also on their fresh properties, such as rheology, shrinkage, hydration, as well as their durability [17].

Despite its much larger effectiveness as compared with microcellulose, confirmed by greater improvements in mechanical properties at lower NC percent addition, the modification of construction materials with nanocellulose is still far from being optimized. In fact, agglomeration phenomena generally start to occur at small additions—0.5–1%, but in some cases even one order of magnitude lower—thus reducing and even canceling the strengthening effect. Because of the relatively low amount of NC added to the mixture, increases in compressive and flexural resistances are generally limited to a few tenths of a percent [11,13,18,19,20,21]. Moreover, the use of superplasticizers in the mixture become critical to ensure a good dispersion of NC and therefore the obtaining of the desired properties [13,21,22,23].

While several works focus on mechanical properties, as previously summarized, much less information is available regarding the modifications that NC produces on durability. Actually, Santos et al. [14] offered an overview of CNF mortar additive ability in enhancing durability. The CNF nanoadmixtures provided cement matrixes with lower sulfate penetration (~50%), greater thermal resistance (no microcracks at 250 °C), and greater resistance to accelerated freeze-thaw (an increase by 20% in flexural strength and 25% in modulus of elasticity after 200 wet-dry cycles). Moreover, the authors focused on CNF-composite preparation that has to comply with specific protocols in the construction sector, first of all homogeneity, so a big effort is put into the development of simplified and efficient dispersion methods for large-scale application of CNFs. 

Indeed, the effects of different CNF dispersion methods on both mechanical and durability properties of cement composites still need to be analyzed in depth. Sonication is the most commonly used method for dispersing NC, although its successful implementation is hampered by two main factors: first, it can disperse NC but can hardly convert NC absorbed on cement particles into free NC, and second, even if NC can be dispersed in water by sonication, more agglomerates may still form in cementitious materials. 

In this work, commercial cotton wool, too short to be spun, was used as an NC source. Although several works indicate high efficiency of bacterial cellulose in reinforcing composites, on account of its high crystallinity, our choice was motivated by both economical and performance-related considerations: indeed, as reported in this work, a high degree of crystallinity is also achievable with a much less expensive starting material, thus more suitable for a future technological transfer. Cotton wool was oxidized to oxidized nanocellulose (ONC) by means of 2,2,6,6-tetramethylpiperidine-1-oxyl (TEMPO)-mediated oxidation [24]. ONC is a charged NC where a certain number of the –CH_2_OH groups are selectively oxidized to the –*COOH* group. Moreover, in a previous study focused on the biomedical field, other chemical modifications were obtained, such as grafting glycidylmethacrylate (GMA) and further cross-linking with ethylene glycol dimethacrylate (EGDMA). The observed increase in mechanical resistance with respect to the starting NC suggested that they can be preferred to BNC for tissue engineering scaffolds in the cases where resistance plays a crucial role [25,26]. For this reason, GMA and EGDMA modifications were also applied in this work, to investigate their potential in cement-based materials strengthening.

The obtained NCs were then used as additives in a commercial mortar, without a super plasticizing agent, and the variation in porosity, mechanical properties as well as in the water-mortar interactions were evaluated, to provide an initial assessment of the obtained material’s durability.

## 2. Materials and Methods

### 2.1. Nanocellulose Production

#### 2.1.1. Oxidized Nanocellulose Preparation (ONC)

Cotton wool was exhaustively milled with FRITSCH GmbH—Milling and Sizing Industriestrasse 8, 55743 Idar-Oberstein. The recovered powder was sifted by a 0.425 mm sieve.

An aqueous solution of TEMPO (Sigma-Aldrich, Milan, Italy, 303 mg) and potassium bromide (KBr, RPE-ACS Analyticals, Carlo Erba, Cornaredo MI, Italy, 2.3 g) was prepared and sonicated for 10 min (400 mL) to ensure the complete solubilization of TEMPO. 

Cellulose (8 g) obtained from milled hydrophilic cotton was soaked into TEMPO/KBr solution for 1 h. A 10% NaClO solution (Supelco, EMPLURA, Milan, Italy 70.0 mL) was added and the reaction proceeded under stirring, while the pH was maintained between 10 and 11 with NaOH (0.5 M). The reaction was concluded when no more pH variations were observed. 

Following acidification with HCl (1 M) to obtain ONC in the acidic form, the sample was filtered on Gooch crucible and thoroughly washed with DI water; EtOH was added in a 2:8 ratio to water to facilitate the following centrifugation step (9000 rpm × 3).

The obtained solid was dispersed in 250 mL DI water. The yield of different preparation ranged from 60 to 70% after lyophilization. 

#### 2.1.2. GMA Grafted ONC

1 g ONC synthesized as indicated in Section 2.1.1. was soaked in warm water (preheated at 80 °C) for 30 min. FeSO_4_∙7H_2_O (Sigma Aldrich, Milan, Italy, 22.2 mg) and H_2_O_2_ (Honeywell, Ph. Eur, Seelze, Germany 710 μL 30%) were added and the suspension was stirred at 80 °C for 25 min. GMA (Sigma-Aldrich, Milan, Italy 2.06 mL) was dropped in the reaction flask, stirring (80 °C) proceeded for 15 min. Excess GMA was removed via hexane extraction and centrifugation (9000 rpm × 3). GMA removal was confirmed by TLC (hexane:AcOEt 1:1). The obtained solid was dispersed in 150 mL DI water. The yield of different preparation ranged from 50 to 60% after lyophilization. 

#### 2.1.3. GMA/EGDMA Grafted ONC

The procedure was conducted on a smaller scale (200 mg to 500 mg ONC). ONC was soaked in warm water (preheated at 80 °C) for 30 min. FeSO_4_∙7H_2_O (0.32 mL of a 0.5 M solution for the 200 mg preparation) (Sigma-Aldrich, Milan, Italy ) and H_2_O_2_ (140 μL for the 200 mg preparation) (Honeywell, Ph. Eur, Seelze, Germany 30%) were added and the suspension was stirred at 80 °C for 25 min at the same concentrations as for the ONC-GMA derivatives. GMA (206 μL for the 200 mg preparation) (Sigma-Aldrich, Milan, Italy) and EGDMA (Sigma-Aldrich) were dropped in the reaction flask, stirring (80 °C) proceeded for 15 min.

The initial synthetic procedure saw a molar ratio between GMA and EGDMA of 1:1. Following GMA:EGDMA investigated ratios were 8:2 and 9:1. Each of the preparations with GMA-EGDMA led to a non-water-dispersible sample and the yield of different preparation ranged from 55 to 60% after lyophilization. 

#### 2.1.4. Suspension

To facilitate the suspension of the different ONCs (with and without GMA or EGDMA), the pH of the suspension was adjusted to 11 with NaOH 0.5 M obtaining the salt form of the carboxylic acid (more hydrophilic). ONCs were further suspended by sonication with an immersed probe at 0 °C (UP100H Ultrasonic Processor, Hielscher, Teltow, Germany), with an output power of 100% (100 Watt ultrasonicator) for one hour (with a stop of a few minutes every 15 min).

The suspension was used for preliminary characterization (DLS, TEM) and for mortar preparation. After ultrasonication, an aliquot was lyophilized to obtain the yield percentage for each reaction, which varied between 60 and 70%, and to conduct IR and SEM analyses.

### 2.2. Mortars Preparation

Mortars were prepared by adding ONC in different percentages to a Portland cement-based premixed commercial mortar Webercem RA30 by Weber (Saint-Gobain Italia, Milan, Italy). The water-to-cement ratio (w/c) was chosen according to indications given for the specific premixed formulation. Table 1 summarizes the content of premixed formulation, water, and nanocellulose of each mortar; nanocellulose percentage is given by weight of cement. A mortar was also produced without nanocellulose, as reference. For each mortar composition, prismatic samples 16 cm × 4 cm × 4 cm were cast.

The manufacture and curing of the mortar specimens were performed according to the standard UNI EN 1015-11:2019 [27]. After cast, mortars were kept at (95 ± 5)% relative humidity, T = 20 °C for 7 days. Samples were then demolded and kept at (65 ± 5)% relative humidity, T = 20 °C for 21 days. At the age of 28 days, the mortar specimens were tested regarding compressive strength [27], as well as to evaluate their interactions with water by capillary absorption tests and contact angle tests.

After the testing campaign, one further formulation was evaluated by adding ONC-GMA to the premix. Given the low solubility and tendency to precipitate of ONC-GMA, to reach a relevant percentage of admixture a large amount of water was necessarily increased to incorporate the solubilization water of nanocellulose. For this reason, mechanical properties were not evaluated in detail, as they would not be comparable with typical w/c ratio conditions.

### 2.3. Characterization Techniques

#### 2.3.1. Infrared Spectroscopy (IR)

Solid phase *FT-IR* spectra of the powdered sample with infrared-grade KBr were collected using an ALPHA spectrometer (Bruker, Bremen, Germany). Data were analyzed using OPUS software, version 7.0 (Bruker, Bremen, Germany). Acquisition of the spectra was performed in the range 4000–400 cm^−1^. 

The oxidation degree (*DO*) of ONC was calculated from the *FT-IR* spectrum as the ratio between the integrated area under the peak of –*COOH* group (from 1830 cm^−1^ to 1670 cm^−1^) and the characteristic peak of cellulose (from 780 cm^−1^ to 480 cm^−1^), see Equation (1). Peak areas were integrated manually.
(1)$$DO_{FT-IR}=\frac{area\;COOH\;\left(manual\;band\;integration\right)}{area\;cellulose\;\left(range\;780-465\;{\mathrm{cm}}^{-1}\;integration\right)}\;\;\;\;\;\;\;\;\;\;\;\;\;\;\;\;\;\;\;\;\;\;$$

The estimation of GMA and EGDMA molar substitution ratio (*MS*) was obtained with the method used in our previous works, see Equation (2), [25,26,28]:(2)$$MS_{FT-IR}=\frac{area\;ester\;\left(manual\;band\;integration\right)}{area\;cellulose\;(range\;780-465\;{\mathrm{cm}}^{-1}\;integration}\;\;\;\;\;\;\;\;\;\;\;\;\;\;\;\;\;\;\;\;\;\;\;\;\;\;\;\;\;\;\;\;\;\;$$

For GMA and EGDMA, the ester peak is the same at 1730 cm^−1^, as stated by literature data and confirmed by recording their *FT-IR*, separately (spectra not reported).

At least two tests were carried out to assure better reproducibility and accuracy for the integration calculations. 

#### 2.3.2. ONC Titration

The ONC potentiometric titration was performed using a NaOH solution (0.1 M). A 0.5% suspension of ONC was used. The degree of oxidation (*DO*) is calculated as the ratio of the added moles of NaOH and the moles of the ONC in suspension [26].

#### 2.3.3. Solid-State ^13^C CP-MAS NMR

^13^C NMR analysis was performed using Dipolar Decoupling Cross Polarization Magic Angle Spinning (DD-CP-MAS) technique with a Bruker Avance 300 spectrometer (Billerica, MA, USA) (75.47 MHz for ^13^C). Concerning data acquisition parameters: repetition time (D1) was equal to 8 s, while contact time and spin rate were 1.6 ms and 10,000 Hz, respectively. 2K scans were collected to obtain good quality spectra. Samples were positioned in a zirconium rotor (diameter: 4 mm; height: 21 mm). Tetramethylsilane was the reference substance for chemical shifts. Benzene was used as a secondary reference standard. The crystallinity index (*Cr. I*%) of nanocellulosic materials was evaluated by means of the following equation used in our previous works, see Equation (3), [25,26,28]:(3)$$Cr.I\;\left(\%\right)=\frac{A}{A+B}\times 100$$
where *A* corresponds to integrals of C4 peaks at 86–92 ppm (crystalline phase) and *B* to the integrals of C4 peaks at 80–86 ppm (other components). The signal components of C6 are evaluated by integrating from 69.0 to 64.5 ppm (crystalline phase) and from 64.5 to 59.3 ppm (other components). The NMR method was chosen to characterize NC crystallinity, as it measures a lower crystallinity index than the XRD method, which underestimates the amorphous content of cellulose [29]. The use of the NMR method was further supported when NC has been extracted from citrus waste [30].

#### 2.3.4. Scanning Electron Microscopy (SEM)

SEM: The surface morphology of samples was observed using a ZEISS EVO-50 EP (ZeissMicroImagingGmbH 2011 CarlZeissNTSGmbHandCarlZeiss, Jena, Germany). This instrument can work at variable pressure (100–120 bar in this case), so the gold coating step could be avoided with a negligible loss of resolution.

#### 2.3.5. Dynamic Light Scattering (DLS)

Hydrodynamic diameter and zeta potential (ζ) values of NC suspensions (0.005%) were measured using Zetasizer Nano ZS (Malvern, Worcestershire, UK) with a fixed 173° scattering angle and a 633-nm helium-neon-laser. Data were analyzed using Zetasizer software, version 7.11 (Malvern, Worcestershire, UK). The temperature was set at 298 K. 

#### 2.3.6. Mortars Performance Characterization

Mortars hardened properties were characterized in terms of mechanical resistance and interactions with water. 

At the age of 28 days, compressive strength was measured on mortar samples (of 4 × 4 × 16 cm) by means of a hydraulic press (Metro Com, KM. 64.300 STRADA ST. 211, Garbagna Novarese, NO 28070, Italy), following the UNI EN 1015-11:2019 standard [27].

Capillary absorption tests were performed following the standard ISO 15148:2002 [31]. Tests were executed by coating 5 of 6 faces of samples with silicone to avoid water evaporation and immersing the lower, non-coated surface in water. Absorption was evaluated by repeatedly weighing specimens at suitable time intervals. The absorption coefficient *A_w_* was evaluated as follows:(4)$$A_{w}=\frac{{\Delta m^{\prime }}_{tf}-{\Delta m^{\prime }}_{0}}{\sqrt{t_{f}}}$$
where Δ*m′_tf_* is mass variation at the time of calculation and Δ*m′*_0_ is the intercept of Δ*m* vs. √*t* at *t* = 0.

Contact angle tests were performed by depositing 50 μL water drops on the surface of the mortars. Five drops per material type were used. Drops were analyzed with the help of the image analysis software ImageJ.

Mortar porosity and pore size distribution were studied by means of mercury intrusion porosimetry (MIP), using a Micromeritics AutoPore IV Automated Mercury Porosimeter (Verder Scientific S.r.l. - Soc. Unipers. Via Pino Longhi, 12, 24066 Pedrengo BG, Italy, pores investigated with radii in the range of 0.003–360 µm). Sample pieces of 1 cm^3^ were oven-dried at 40 ± 5 °C for 8 h prior to the analysis.

Finally, thermal gravimetric analyses (TGA) were conducted on ONC-modified mortars and on the reference material, to assess possible variations in the thermal stability of the materials. Mortar samples of some tens of grams were crushed and powders were accurately mixed, then two small samples per each mortar were used for testing: this procedure allows minimizing possible variabilities of mortar composition from one point to another, homogenizing its composition. Tests were performed with a Perkin Elmer STA 6000 (Milano, Italy); a temperature ramp of 10 °C/min was applied in air in a temperature range 10–900 °C.

## 3. Results

### 3.1. Cotton Wool Characterization 

Figure 1 shows data related to cotton wool: *FT-IR*, as a pure cellulose, and SEM, where its fibers are clearly visible and their thickness can be evaluated in the order of magnitude of tens of micrometers.

### 3.2. Nanocellulose Characterization

#### 3.2.1. FT-IR Analyses

*FT-IR* analyses were conducted on the acid form of oxidized nanocellulose (ONC), see Figure 2, and the *DO* was calculated to be ranging from 0.31 to 0.40, according to Section 2.3.1 and as reported in the figure.

Two ONC *FT-IR* absorbance spectra are further displayed in Figure 3, depending on the *COOH* group, in the acidic form (green) or in the salt form (blue). The characteristic band of carboxylate anion is around 1600 cm^−1^, overlapping with the signal of bound water of cellulose (blu). The band at 1729 cm^−1^ is descriptive of the *COOH* group (blue). Other bands representative of the *COOH* group (between 1440 and 1375 cm^−1^) are negligible due to overlapping with other stronger characteristic signals of cellulose. 

ONC was also titrated by potentiometric measures, as detailed in Section 2.3.2. *DO* was obtained by the ratio of the moles of NaOH used for the titration with the moles of the ONC in suspension. ONC’s *DO* by titration is quite similar to the *DO* obtained with *FT-IR*, data not reported.

The ONC degree of oxidation (*DO*) is also evaluated by ^13^C CP-MAS NMR spectra (see Section 3.2.3). Signals of primary alcohols are characterized by a greater quantitative response than the signals from secondary alcohols. C6 acid groups obtained by oxidation are present in the order of 27%.

*FT-IR* analyses on ONC-GMA and ONC-GMA/EGDMA were conducted on the salt form in order to avoid the overlapping of C=O stretch signals (acidic form) with the ones specific to GMA and EGDMA (ester group, 1730 cm^−1^). The *FT-IR* spectrum of ONC-GMA shows the typical cellulose bands and the additional characteristic peak of glycidyl methacrylate at 1730 cm^−1^. From such spectrum, a *MS* of 0.38 was calculated (Figure 4a).

In Figure 4b, the spectrum of the ONC salt form (red) is compared to the ONC-GMA-salt form spectrum (blue). The characteristic peak of the –COO^−^ group overlaps with the characteristic peak of bound water. This means that functionalization of the ONC is obtained. It is possible to calculate the molar substitution ratio (*MS*) taking into account the ratio between the area under the peak of the GMA at 1730 cm^−1^ (units of grafted GMA) and the area under the characteristic peak of cellulose (units of anhydrous D-glucose). 

ONC-GMA-EGMA *FT-IR* spectrum is reported in Figure 5. As expected, the spectrum is quite similar to ONC-GMA spectra, as the ester group is the same. The molar substitution ratio is 0.52 for ONC-GMA/EGDMA.

#### 3.2.2. DLS Analyses

Suspensions were diluted 1:10 to conduct DLS analyses to confirm the nanoscale dimension of final samples.

DLS analysis of synthesized ONC samples in the basic form confirmed the downsizing of fibers to nanometer scale, whereas ONC-GMA samples were found to have a higher Z-average (avg diameter) and polydispersity index due to the GMA hydrophobic appendages that favor aggregation phenomena (Figure 6, Table 2). DLS analysis on GMA/EGDMA grafted ONC was not conducted due to the impossibility of obtaining a homogeneous suspension.

Analyses of the Z-Potential showed a neat negative superficial charge (Table 2) on both ONC and ONC-GMA samples due to the presence of carboxylic acid moiety on C6.

As reference, the native cellulose nano fibrils ζ at pH = 7 was found to be −35 mV [32]. As expected, TEMPO oxidation makes NC ζ more negative. Functionalization with GMA makes ONC-GMA ζ less negative. The steric hindrance of the GMA networks may provoke a decrease at the level of the surface charge of functional groups of ONC.

#### 3.2.3. NMR Analyses

NMR analyses were conducted on starting material (cellulose from cotton wool), milled cellulose and on the obtained nanocellulose in order to better understand how the process of milling and oxidation influences the crystallinity of cellulose. An allomorph cellulose 1 at 40% of the sample mass was used as the starting material. 

NMR results (Figure 7) indicate that mechanical treatments do not involve changes in the allomorphic form of the sample, but they do induce a loss of the less structured components in the order of 18%, increasing the percentage of crystalline phase up to 58% as measured on C4 signals. In other words, milling has an impact on cellulose crystallinity and specifically, it mostly breaks cellulose fibers in their amorphous areas, thus leading to an overall increase in crystallinity. The subsequent oxidative process increases the crystalline phase up to a value of 67%, while maintaining the allomorphic state of cellulose 1.

Spectra were processed and two series of crystallinity indexes were obtained, one referring to primary alcohols and the other to secondary alcohol (Table 3) [33]. In the ^13^C CP-MAS NMR spectra, signals of primary alcohols are characterized by a greater quantitative response than signals of secondary alcohols. With the parameters used for our measurements, this value is in the order of 12%. Having made this premise, from the table it can be deduced from the total recovery of C6 signal with respect to the anomeric signal that in ONC acid groups are of the order of 27%, thus confirming *FT-IR* results.

#### 3.2.4. SEM analyses

SEM analyses of lyophilized ONC, ONC-GMA and on selected mortars are reported in Figure 8. 

ONC morphology (Figure 8A) suggests the formation of an aerogel, as also reported in previous works [34]. On the other hand, ONC-GMA apparently produces a more continuous structure, as a consequence of its polymerization (Figure 8B).

Concerning mortars, when 0.6% ONC is added, the presence of nanocellulose fibrils is evident, as opposed to the reference mortar (without NC addition, Figure 8C). Small aggregates are indeed observed, as highlighted in Figure 8D. Such agglomerates were not found in the mortar containing 2.4% ONC, likely because of a better dispersion in the mixture (Figure 8E). 

### 3.3. Mortars Characterization

#### 3.3.1. Mechanical Properties

First, mortars were tested for compressive resistance, Rc. Results are reported in Figure 9a. The addition of nanocellulose in the form of ONC leads to an increase in compressive resistance. Error bars were not included in the graph as results were very repeatable (avg error: 2%, max error: 3%). At low percentages of nanocellulose (0.3% and 0.6% ONC by weight of cement) no significant improvement is noticed with respect to the standard mortar, although at 0.6% a slight increase in resistance starts to appear. This trend is confirmed and amplified at higher contents of ONC.

Results become clearer if observed in terms of %Rc increase as a function of ONC content (Figure 9b). Sample ONC-0.3 is not included, as that ONC content was not sufficient to impart an increase in Rc to the mixture. Indeed, even at 0.6% ONC the effect on Rc is clearly visible, if compared with the low amount of nanocellulose introduced. A quasi-linear increase in Rc can be identified with increasing ONC addition, indicating the strengthening effect is strongly dependent on the additive quantity: indeed, the slope of the interpolating linear trend indicates approximately 34% increase in Rc per unit % of ONC added to the mixture. 

#### 3.3.2. Water Absorption and Contact Angle

Capillary absorption tests (Figure 10) indicate a reduction in capillary absorption with respect to the reference material starting from a percentage of ONC of 0.6%. The increase in absorption at 0.3% ONC is coherent with mechanical resistance tests, where a slight decrease in Rc was observed. Once a minimum ONC content is reached, the absorption alteration is not related with its concentration: overall, nanocellulose addition was able to decrease the water absorption coefficient A_w_ by approximately 25%.

Contact angle measurements did not allow observing relevant differences among ONC-modified mortars and standard ones; all mortars showed a contact angle of approximately 95° (Figure 11). Hence, the addition of ONC did not affect this parameter in the hydrophilic direction nor in the hydrophobic one. For this reason, we can state that differences observed in water absorption shall be assigned completely to alterations in pore size and structure, since chemistry was not relevantly affected as far as surface tension is concerned.

#### 3.3.3. Porosity Tests

Figure 12 summarizes pore size distribution profiles of all samples tested: the addition of small quantities of ONC seems to have only limited effect on the mortar microstructure, but when 1.2% and 2.4% ONC were added, the microstructure appeared strongly modified, with a sharp decrease in pore size as well as in the overall value of incremental intrusion. This information is coherent with the increase in mechanical resistance and water absorption observed in previously described tests. The increased porosity at 0.3% ONC explains the slight decrease in mechanical resistance and increase in water absorption observed on this formulation.

Data were then processed to obtain quantitative information (Figure 13). Both total porosity and median pore size decrease with the addition of 1.2% and 2.4% nanocellulose, confirming not only a reduction in porosity—which can be ascribed to a larger hydration degree, coherent with the increased mechanical resistance observed and with previous literature [20]—but also an effective pore segmentation, which is beneficial not only to mechanical resistance, but also to the material durability [35]. In this direction, a further improvement is provided by the increase in micropores (<50 nm) present in the material. it is well-known that micropores are not involved in the transport of substances inside cement paste porosities: therefore, the larger their contribution to the overall porosity, which is a further evidence of pore segmentation, the better the material durability [35]. Indeed, the addition of 1.2% NC or more increased by more than 50% the percent weight of micropores on total porosity, passing from 10% of the reference material to 17% of the mortar with higher NC addition.

#### 3.3.4. Thermal gravimetric analyses (TGA)

Figure 14 reports both percent weight variation and its derivative as a function of temperature for two samples, the reference mortar (no NC) and the one with the largest modification (2.4% ONC). No relevant modification was observed in the presence of ONC, an exception made for a reduction in the residual mass at the maximum test temperature. The latter is related to the complete combustion of ONC, and therefore an overall decrease in the residual unburned material (minerals). However, no peak shift or evident change in their intensity is noticed, indicating that the material thermal stability is not affected by the presence of ONC. Tests were repeated on all samples, but analogous results were obtained (data not shown for better image clarity).

## 4. Discussion

Starting this discussion by bringing attention to the old story of cellulose and its acrylates derivatives is, in our opinion, useful to understand the rationale of this research of preparing a range of different NCs, from which to select a suitable additive for mortar [36]. The functionalization of cellulose in solid state with glycidyl methacrylate by chemical and physical methods, and subsequent glycidyl modification, led to different solid cellulose derivatives that merge cellulose property with new properties induced by the functionalization. Cellulose-based materials for sanitary and environmental fields were published and patented, as exhaustively summarized in [36]. The rationale was in any case to perform solid-state cellulose surface activation with the aim of inducing new properties depending on the activation type and level. 

In this study, the objective was to investigate different functionalization methods, which proved effective in previous works focused on other applications, and finally selecting the NC most suitable to be developed as mortar additive. For this reason, nanocellulose chemistry and characterization is propaedeutic to this research. To meet the final aim, i.e., use NC as additive in mortars, a further objective was to obtain NC suspensions, to obtain mortars with peculiar properties and avoid NC agglomeration or uneven dispersion. Eventually, NCs prepared can also be considered for other applications.

For this research, cotton wool was reduced to ONC by partial selective CH_2_OH oxidation, as summarized in Figure 15. Cotton wool is a well-known sanitary industrial product, recovered as silky fibers from cotton plants in their raw state, after seeds removal and bleaching process using hydrogen peroxide or sodium hypochlorite. Indeed, no pulping process was necessary before the extraction of ONC from cotton wool. Similarly, in a previous work, a high-grade pure industrial cotton waste fiber, too short to be spun, was used as NC source: this demonstrate the feasibility of exploiting an industrial, clean waste as source for the production of NC additives of interest to the cement industry [26].

A trivial but relevant consideration is that the richer the starting material with respect to cellulose, the less important the pre-treatment before cellulose reduction to nano size. Actually, nanocellulose recovery from textile waste is a concrete opportunity to take advantage of cellulose-containing waste and to enhance it [37]. 

ONC is a flexible material that can be used in acid form, which means operating with a charged material—although acidity is weak—or in salt form, which means operating with an ionic material. At the same time, ONC maintains the well-known peculiar cellulose properties. The ONC’s structure makes it particularly suitable to be suspended in water and endows ONC suspension with easier interaction with cement constituents during hydration. 

ONC was further functionalized by GMA and EGDMA, as described in our previous work and specified in its Scheme 1 [25], creating ONC-GMA and ONC-GMA/EGDMA. Although ONC-GMA was successfully suspended in water, but at lower concentration than ONC, its application resulted in being not useful in the field of cement-based materials, because of the exceeding quantity of water needed to suspend it and that would adversely affect the material microstructure. ONC-GMA/EGDMA was not suitable to be suspended, hence it was discarded as well. ONC-GMA behavior toward suspension can be explained by the hydrophobicity of GMA appendages, while for ONC-GMA/EGDMA the hydrophobic EGDMA reticulation, even in low amount, hinders the suspension. Yet, ONC-GMA and ONC-GMA/EGDMA are multifunctional nanomaterials that can preview different applications, not only as it is, but also by modifying the epoxide ring of the glycidyl residue, in agreement with fundamental organic chemistry notions.

As far as applications as additive in cement-based materials are concerned, ONC is the only material that has actually shown a promising behavior, as the possibility of suspending it up to relevant concentrations allowed its use as admixture without impacting on the mortar water-to-cement ratio. Comparing the materials developed in this research with previous works, a larger quantity of nanocellulose was admixed with no flocculation issues, with consequent larger increases in compressive strength with respect to what is generally achieved. Indeed, as summarized by Santos et al. ([14] and references therein), in most works where NC reaches significant % additions it is added as powder in the mixer and a plasticizer is required: this result is a lower improvement in mechanical resistance. Our choice of suspending NC clearly improved dispersion, making its presence more effective as also recently shown by Oliveira de Souza et al. [23].

The increase in strength is a direct consequence of a decreased porosity, as can be observed from compressive strength and capillary absorption trends as a function of material porosity reported in Figure 16: indeed, both properties show a linear trend with porosity. This trend also confirms, as previously hypothesized that capillary absorption is mostly related with pore size and overall porosity rather than with changes in chemical composition and therefore surface tension, as proved by the unaltered contact angle. Moreover, and most notably, the decreased absorptivity, decrease in pore size and increase in quantity of micropores are indicative of an improved material durability, which is as much relevant as its strengthening for applications as mortar additive as summarized-again-by Santos et al. ([14] and references therein). These results are therefore a starting point to investigate the durability of ONC-containing mortars.

## 5. Conclusions

In conclusion, this research work showed the feasibility of producing nanocellulose from a clean scrap material of industrial production, and of modifying its structure to introduce functionalities that can improve its dispersion in cement-based matrixes. Nanocellulose was produced in thin filaments. The nanocellulose admixture was successfully added in relatively high quantity to a commercial mortar, modifying its hardened properties. An increase in compressive resistance of more than 100% was achieved, and water absorption decreased by 30%. Both modifications were ensured by the improved pore segmentation, as attested by the decreased overall porosity, smaller pore size and increase in micropores percentage. These features are indicative of an improved material durability.

## Figures and Tables

**Figure 1 nanomaterials-12-01093-f001:**
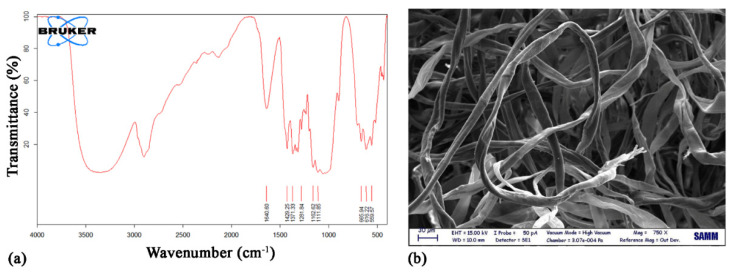
(**a**) *FT-IR* and (**b**) SEM analyses of cotton wool.

**Figure 2 nanomaterials-12-01093-f002:**
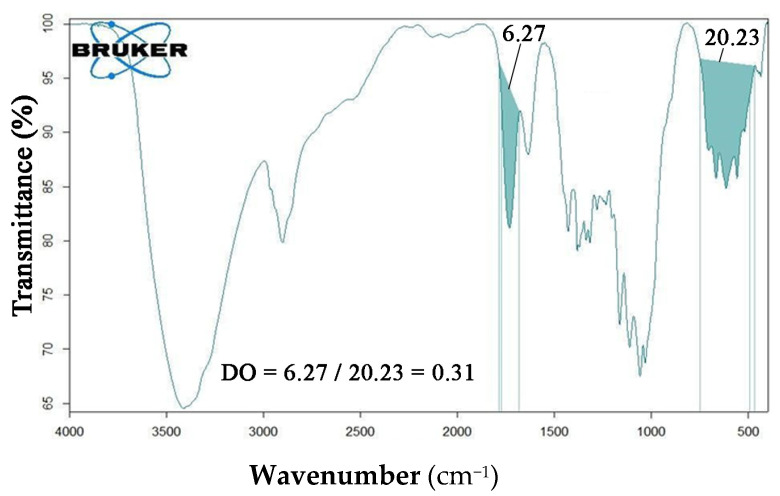
*FT-IR* analysis of ONC in the acid form.

**Figure 3 nanomaterials-12-01093-f003:**
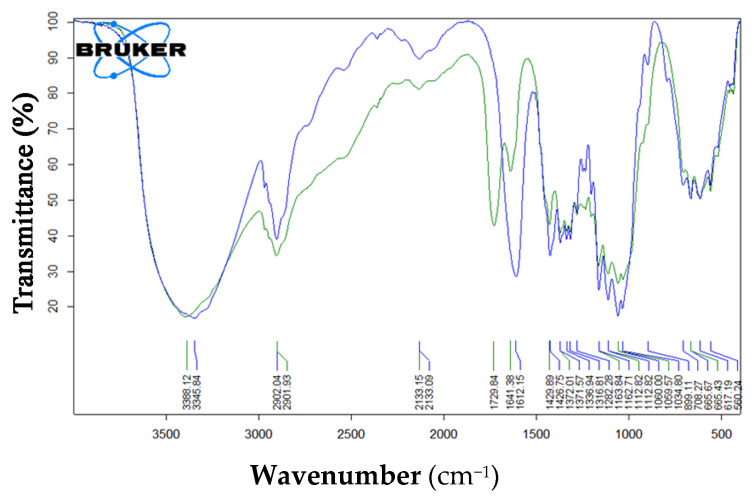
*FT-IR* spectra of ONC in its acidic (green) and salt (blue) forms.

**Figure 4 nanomaterials-12-01093-f004:**
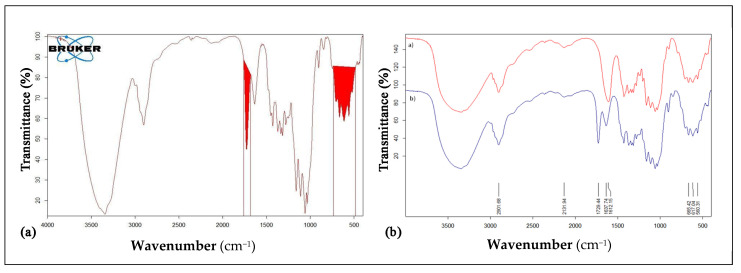
(**a**) FT-IT analysis of ONC-GMA in salt form highlighting peaks integrated for *MS* calculations. (**b**) Comparison between *FT-IR* spectra of ONC-GMA in its salt form (blue) and ONC salt form (red).

**Figure 5 nanomaterials-12-01093-f005:**
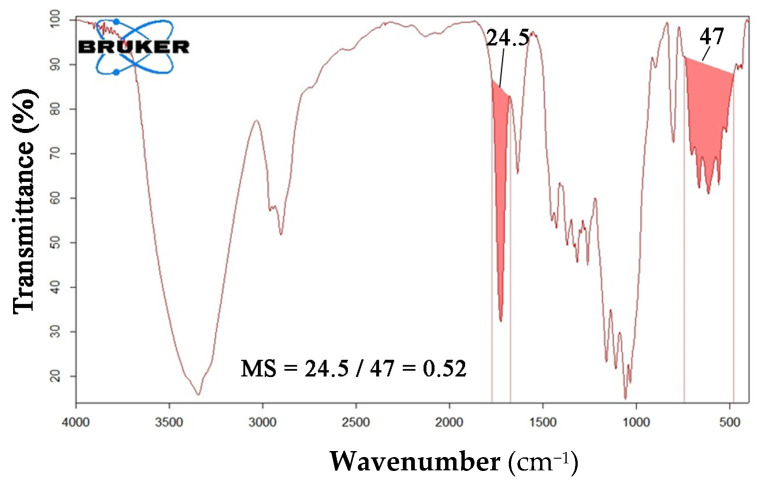
*FT-IR* analysis of ONC-GMA/EGDMA.

**Figure 6 nanomaterials-12-01093-f006:**
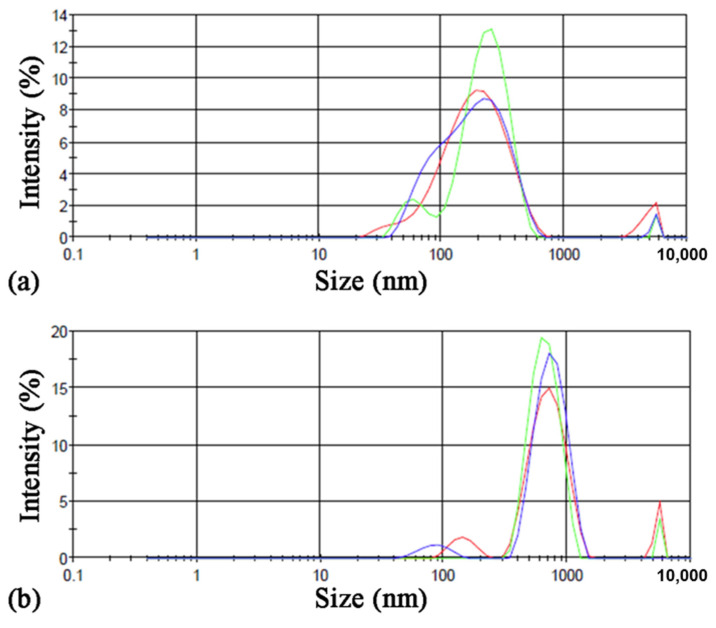
DLS size analysis of (**a**) synthesized ONC and (**b**) synthesized ONC-GMA.

**Figure 7 nanomaterials-12-01093-f007:**
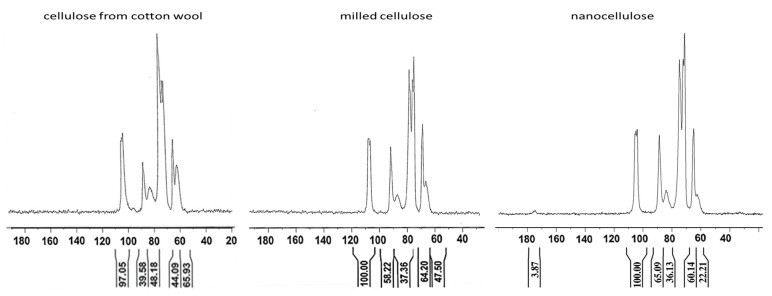
^13^C CP-MAS NMR spectra of cellulose from cotton wool, milled cellulose and nanocellulose (ONC).

**Figure 8 nanomaterials-12-01093-f008:**
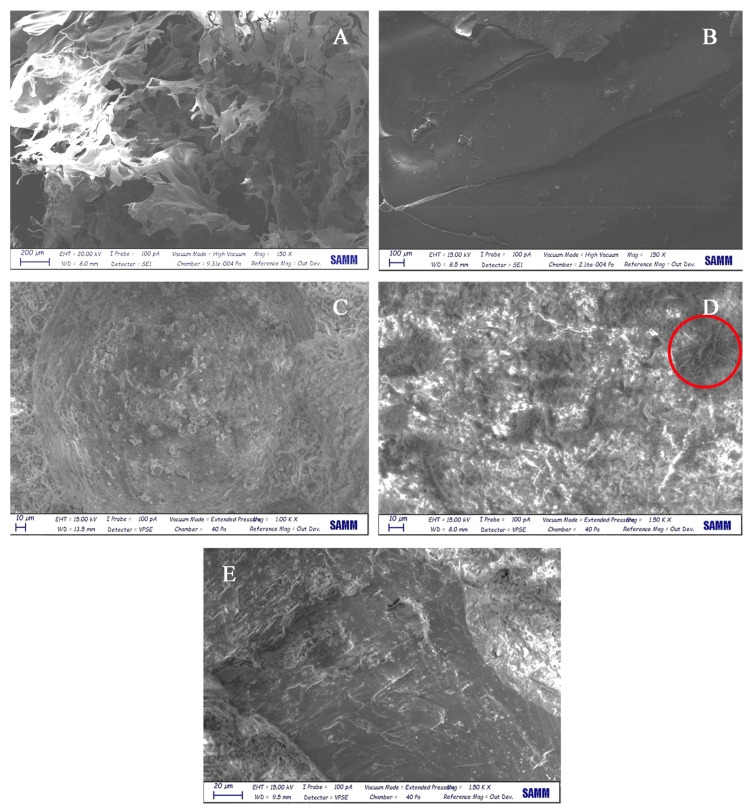
SEM analyses. (**A**) Lyophilized ONC; (**B**) lyophilized ONC-GMA; (**C**) reference mortar (0% ONC); (**D**) mortar with 0.6% ONC; (**E**) mortar with 2.4% ONC.

**Figure 9 nanomaterials-12-01093-f009:**
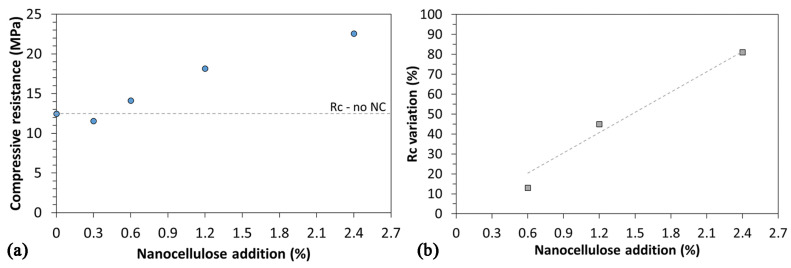
(**a**) Compressive resistance Rc of mortars as a function of ONC content; (**b**) Rc variation of mortars as a function of ONC content.

**Figure 10 nanomaterials-12-01093-f010:**
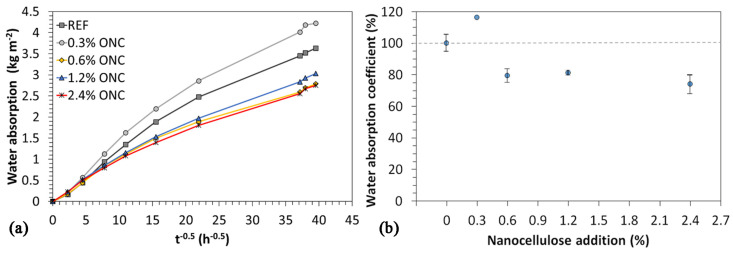
Measured water absorption trends vs. time of mortars with different ONC content (average profiles over three samples per type).

**Figure 11 nanomaterials-12-01093-f011:**
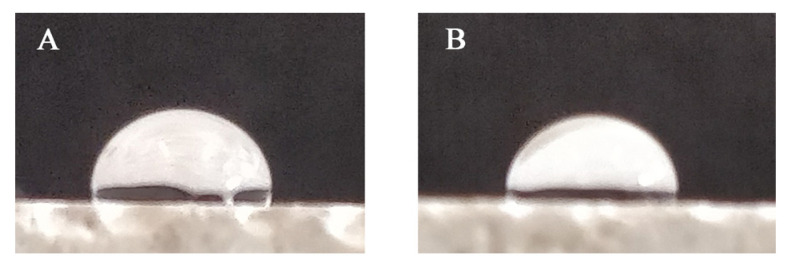
Contact angle measurement on reference mortar (**A**) and on a mortar containing 1.2% ONC (**B**).

**Figure 12 nanomaterials-12-01093-f012:**
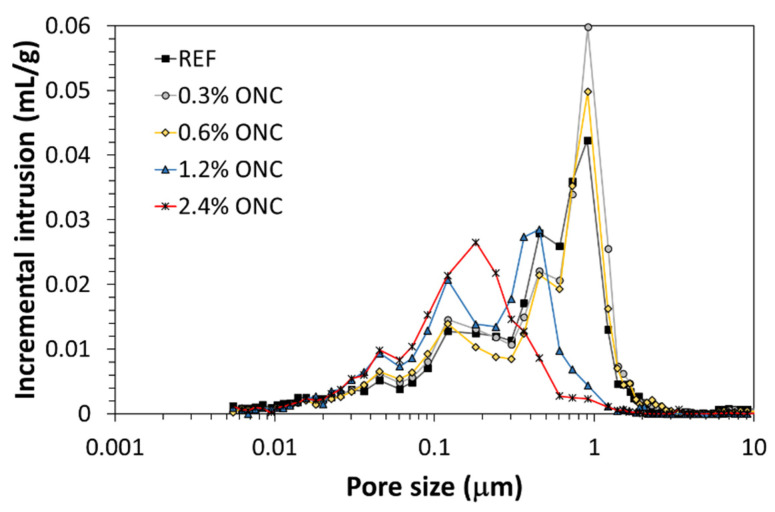
Pore size distribution of mortars with different ONC content.

**Figure 13 nanomaterials-12-01093-f013:**
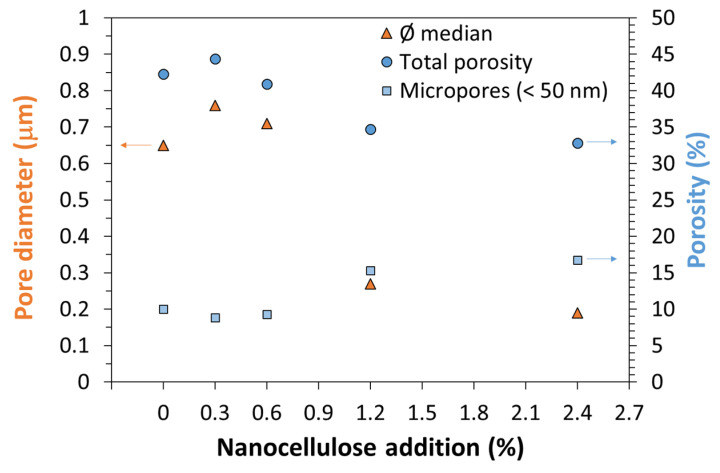
Summary of porosity data of mortars with different ONC content: median diameter, total porosity and micropores percentage are reported as representative of changes in pore structure.

**Figure 14 nanomaterials-12-01093-f014:**
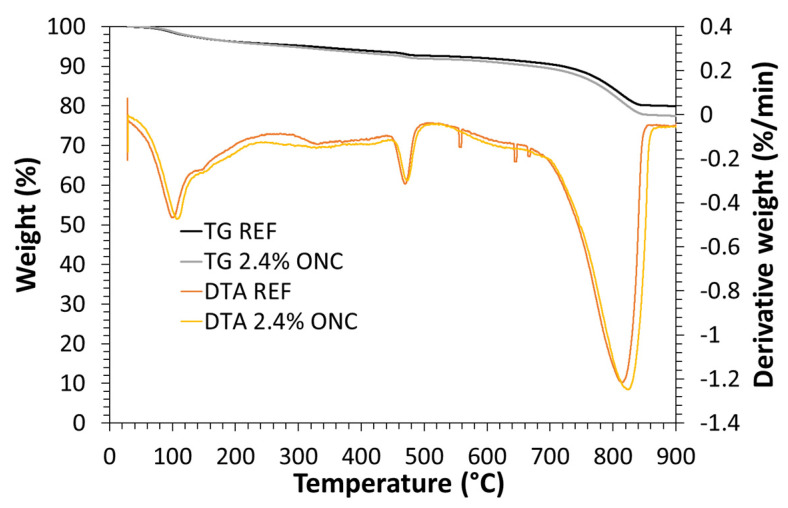
TGA measurements reporting samples percent weight and derivative weight for standard mortars without NC and a mortar with 2.4% ONC.

**Figure 15 nanomaterials-12-01093-f015:**
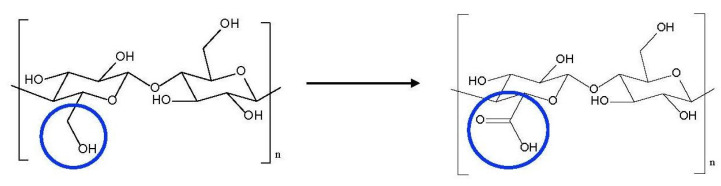
TEMPO mediated cellulose oxidation.

**Figure 16 nanomaterials-12-01093-f016:**
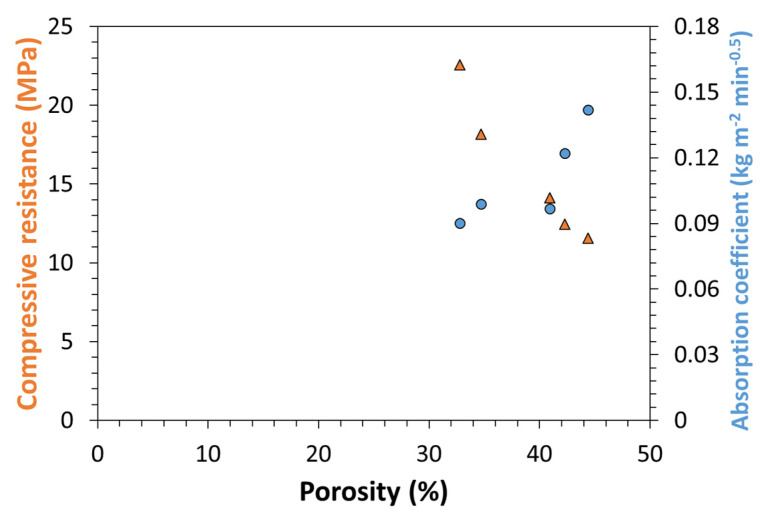
Compressive strength and absorption coefficient versus material total porosity.

**Table 1 nanomaterials-12-01093-t001:** Mixture proportion for mortar samples preparation.

	Water (L/m^3^)	Premix (kg/m^3^)	w/c	ONC (kg/m^3^)	ONC (%)
REF	410	1700	0.48	-	-
ONC-0.3	410	1700	0.48	2.6	0.3
ONC-0.6	410	1700	0.48	5.1	0.6
ONC-1.2	410	1700	0.48	10.2	1.2
ONC-2.4	410	1700	0.48	20.4	2.4

**Table 2 nanomaterials-12-01093-t002:** DLS results for ONC and ONC-GMA.

	Z-Average (d, nm)	PdI	Intercept	Z-Potential (mV)
ONC	167.7	0.448	0.835	−38.4
ONC-GMA	761.0	0.696	0.824	−33.6

**Table 3 nanomaterials-12-01093-t003:** Integral % values with respect to anomeric signals from 1^3^C NMR spectra.

	C4 Crystalline *	C6 Crystalline *
Cotton cellulose	40.7 (90.7)	45.4 (113)
Milled cellulose	58 (95.6)	64 (111.7)
ONC	65 (101.22)	60 (82)

* In bracket the total % signal area recovery of crystalline and non-crystalline phases.

## Data Availability

Data are contained within the article.

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
