# Peer review of "Suspended Multifunctional Nanocellulose as Additive for Mortars"

_nanomaterials, 2022, doi:10.3390/nano12071093_

Round 1

Reviewer 1 Report

Maria Vittoria Diamanti et al. prepared a charged nanocellulose capable of electrostatic interaction, merging cellulose and nanoparticles properties with high-purity cotton wool as raw material, and then oxidized it to produce oxidized nanocellulose (ONC). ONC was further functionalized with the acrylate branch bound to the cellulose skeleton by C-C bonds, thus maintaining the OH groups intact. The effect of the modified microstructure of ONC on mortar mechanical properties and water-mortar interaction was evaluated by adding ONC into a commercial mortar. It was found that the addition of ONC increased the compressive resistance of the mortar by more than 100% and reduced the water absorption by 30%. Meanwhile, the addition of ONC caused an alteration of mortar porosity, with a decreased percent porosity and pore size distribution shifted towards smaller pores, with consequent increase in the percentage of micropores present in the material, indicating a potential improvement in mortar durability. These results can provide some references for the use of modified nanocellulose as an additive in cement-based materials, but there are still many problems in the paper, which must be major revised before it can be accepted.

Introduction

  1. Page 2, “Only in very recent articles, starting 2019, some research works also analyzed the behavior of NC-modified mortars and concrete to phenomena such as freeze-thaw, sulphate attack, carbonation and chlorides penetration”, please briefly describe the current strategy of NC-modified mortars.

Materials and Methods

  1. The authors did not provide a number of important characteristics of the high-grade pure industrial cotton (cotton wool).
  2. The authors did not provide “Statistical Analysis” section.

Results

  1. Please supplement the unoxidized NC in the DLS results to compare with ONC and ONC-GMA.
  2. The effect of ONC-GMA and ONCGMA/EGDMA on mortar properties has not been studied, so what is the point of doing these two? Data on ONC-GMA and ONCGMA/EGDMA are also often lacking when characterizing ONC properties and need to be supplemented.
  3. XRD of ONC, ONC-GMA and ONCGMA/EGDMA should be provided to better understand how the process of milling and oxidation impacts the crystallinity of cellulose.
  4. There is a lack of AFM results that more intuitively represent roughness in mortar characterization, please add.
  5. What is the effect of nanocellulose on the thermal properties of mortar? Please supplement the TGA, DTG or DSC results.
  6. The results and analysis of all performances only simply stated the results, without analysis and discussion of the reasons for the results, or comparison with relevant literature, which could not reflect the advantages.
  7. Discussion section, “ONC-GMA/EGDMA was not suitable to be suspended, hence it was discarded as well.” Why is it not suitable for suspension? Reasonable explanations should be given and supplemented by relevant experimental data. Since it is not suitable for suspension, what is the reason for doing this?
  8. Please check the full text for formatting and spelling.

References

  1. The authors could add the following references which would again increase the interest to general functional cellulosic material readers: Journal of Bioresources and Bioproducts, 2021, 6(1): 26-32; ACS Applied Materials & Interfaces, 2021, 13, 7617-7624; Journal of Bioresources and Bioproducts, 2021, 6(1): 75-81.

Author Response

Response to reviewers

We thank the reviewers for the points raised, which helped us making the article clearer to readers and more scientifically sound.

Below you can find our specific answers to each single consideration.

REVIEWER 1

Maria Vittoria Diamanti et al. prepared a charged nanocellulose capable of electrostatic interaction, merging cellulose and nanoparticles properties with high-purity cotton wool as raw material, and then oxidized it to produce oxidized nanocellulose (ONC). ONC was further functionalized with the acrylate branch bound to the cellulose skeleton by C-C bonds, thus maintaining the OH groups intact. The effect of the modified microstructure of ONC on mortar mechanical properties and water-mortar interaction was evaluated by adding ONC into a commercial mortar. It was found that the addition of ONC increased the compressive resistance of the mortar by more than 100% and reduced the water absorption by 30%. Meanwhile, the addition of ONC caused an alteration of mortar porosity, with a decreased percent porosity and pore size distribution shifted towards smaller pores, with consequent increase in the percentage of micropores present in the material, indicating a potential improvement in mortar durability. These results can provide some references for the use of modified nanocellulose as an additive in cement-based materials, but there are still many problems in the paper, which must be major revised before it can be accepted.

Introduction

  1. Page 2, “Only in very recent articles, starting 2019, some research works also analyzed the behavior of NC-modified mortars and concrete to phenomena such as freeze-thaw, sulphate attack, carbonation and chlorides penetration”, please briefly describe the current strategy of NC-modified mortars.

We thank the reviewer for pointing out this weakness in the introduction. We have now implemented a better analysis of this literature section.

Materials and Methods

  1. The authors did not provide a number of important characteristics of the high-grade pure industrial cotton (cotton wool).

We apologize for such missing data, we have now added FT-IR, SEM and NMR of the cotton wool used in the study.

  1. The authors did not provide “Statistical Analysis” section.

A statistical analysis section is not provided since only a small number of samples was analyzed in each test, therefore there was no possible statistical approach that could be applied.

Results

  1. Please supplement the unoxidized NC in the DLS results to compare with ONC and ONC-GMA.

We agree these data were also needed. They have now been added to the paragraph 3.3.4.

  1. The effect of ONC-GMA and ONCGMA/EGDMA on mortar properties has not been studied, so what is the point of doing these two? Data on ONC-GMA and ONCGMA/EGDMA are also often lacking when characterizing ONC properties and need to be supplemented.

We apologize for the lack of clarity of this section. As now better specified, these two modifications of ONC were tested as they proved very interesting in previous works related to the biomedical field. Unfortunately, in this study we didn’t obtain a sufficient solubility, therefore we could not add them to the mortar without significantly altering its w/c ratio. Still, we believe this result needs to be presented. We have now modified both the introduction final part and the results description related to these two ONC modifications to better explain why we tested them, and what went wrong.

  1. XRD of ONC, ONC-GMA and ONCGMA/EGDMA should be provided to better understand how the process of milling and oxidation impacts the crystallinity of cellulose.

We have performed NMR measurements to provide information on crystallinity and its alteration as a consequence of milling and oxidation. Results are reported in par 3.2.3. We firmly believe that NMR is a better characterization technique in this sense, as it has higher sensitivity compared with XRD. To underline this, we have implemented the presentation of this technique in the experimental section and added some comments on results.

  1. There is a lack of AFM results that more intuitively represent roughness in mortar characterization, please add.

Undoubtedly, AFM could provide information on roughness, yet, here we are not addressing mortars roughness but rather their porosity, which is a key parameter to understand their mechanical properties and their durability, which instead do not depend on roughness. For this reason we chose to analyze porosity by mercury intrusion porosimetry.

  1. What is the effect of nanocellulose on the thermal properties of mortar? Please supplement the TGA, DTG or DSC results.

We thank the reviewer for the suggestion. These tests have now been added to the work (Par. 3.3.4). Unfortunately, no big influence of ONC was observed on TGA results.

  1. The results and analysis of all performances only simply stated the results, without analysis and discussion of the reasons for the results, or comparison with relevant literature, which could not reflect the advantages.

We apologize for the lack of discussion, we have now improved the discussion section

  1. Discussion section, “ONC-GMA/EGDMA was not suitable to be suspended, hence it was discarded as well.” Why is it not suitable for suspension? Reasonable explanations should be given and supplemented by relevant experimental data. Since it is not suitable for suspension, what is the reason for doing this?
  2. Please check the full text for formatting and spelling.

The text was fully proofread.

References

  1. The authors could add the following references which would again increase the interest to general functional cellulosic material readers: Journal of Bioresources and Bioproducts, 2021, 6(1): 26-32; ACS Applied Materials & Interfaces, 2021, 13, 7617-7624; Journal of Bioresources and Bioproducts, 2021, 6(1): 75-81.

We thank the reviewer for the suggestion, we have added the suggested articles.

Reviewer 2 Report

An interesting application of nanocellulose which progresses the topic beyond previous studies

Specific observations

  1. The authors define ONC  in the abstract  but it is only used once thereafter in the abstract, It would be simpler to spell out ONC both times it is used  in the abstract
  2. In this case ONC is used without definition on page 2. The Instructions for Authors make it clear that any acronym must be defined the first time it is used in the abstract, the main text and a figure caption
  3. P4 the authors described a method for quantitative analysis of FTIR spectra but the text does not make it clear how this was done. The figures on p6 and 7 show transmission spectra . Moreover I take it the coloured areas correspond to the area of the peak but these are only a part of the spectra that encloses this area. How were these areas obtained? and Why work in Transmission spectra
  4. The authors use nmr to evaluate the fraction of crystallinity  and the form of the cellulose crystal structure ie 1 or 2  etc  No information is given as to how this is peformed
  5. With reference to 3, the authors give two references, the first I could access but this simply repeats the same information as in this ms. However a new reference is gen number 19 in the paper ref 22 but this reference describes the use of X-ray scattering to determine crystallinity and the crystal structure type. The authors must decribe the procedures used to allow others to follow their work
  6. The text in the line at the bottom of page 10 2 to 2Onm conflicts with the values given in the image in Figure 8 and in the conclusions.
  7. SEM Images Figure 7  These are very low contrast images especially A, B and C.  I note that images C, D and E were obtained  using the variable pressure mode of  the SEM whereas A and B were obtained under high vacuum conditions this seems in conflict with the text of page 5 
  8.  These areas must be addressed by the authors to provide an adequate description of their work

Author Response

Response to reviewers

We thank the reviewers for the points raised, which helped us making the article clearer to readers and more scientifically sound.

Below you can find our specific answers to each single consideration.

REVIEWER 2

An interesting application of nanocellulose, which progresses the topic beyond previous studies

Specific observations

  1. The authors define ONC  in the abstract  but it is only used once thereafter in the abstract, It would be simpler to spell out ONC both times it is used  in the abstract
  2. In this case ONC is used without definition on page 2. The Instructions for Authors make it clear that any acronym must be defined the first time it is used in the abstract, the main text and a figure caption

We thank the reviewer for the suggestion, we have modified the abstract and added the acronym to the main text accordingly.

  1. P4 the authors described a method for quantitative analysis of FTIR spectra but the text does not make it clear how this was done. The figures on p6 and 7 show transmission spectra . Moreover I take it the coloured areas correspond to the area of the peak but these are only a part of the spectra that encloses this area. How were these areas obtained? and Why work in Transmission spectra
  2. With reference to 3, the authors give two references, the first I could access but this simply repeats the same information as in this ms. However a new reference is gen number 19 in the paper ref 22 but this reference describes the use of X-ray scattering to determine crystallinity and the crystal structure type. The authors must decribe the procedures used to allow others to follow their work

We apologize for the lack of clarity. FT-IR measurements are described in par. 2.3.1, indicating that measurements were performed on KBr-NC solid tablets. IR signal can be analyzed and integrated both in absorbance and transmittance mode; we decided to use the transmittance mode as the difference between the two types of signal was negligible. The calculation of oxidation degree e molar substitution from FTIR measurements is explained with related equations in par 2.3.1, and in the equation it is stated that peaks were manually integrated. We have now specified this also in the main text and in the figures. References related to this section refer to the calculation of molar substitution fraction, not to crystallinity. Moreover, a more correct reference is now cited in this section.

  1. The authors use nmr to evaluate the fraction of crystallinity  and the form of the cellulose crystal structure ie 1 or 2  etc  No information is given as to how this is peformed

Again we apologize for the lack of clarity. Actually, also in this case the calculation of crystallinity is explained with related equations in par 2.3.2, and reference is made to previous works where this method was previously used and explained.

  1. The text in the line at the bottom of page 10 2 to 20 nm conflicts with the values given in the image in Figure 8 and in the conclusions.

We are sorry for the mistake in reading values on the image, the values have now been amended.

  1. SEM Images Figure 7  These are very low contrast images especially A, B and C.  I note that images C, D and E were obtained  using the variable pressure mode of  the SEM whereas A and B were obtained under high vacuum conditions this seems in conflict with the text of page 5 

We have fixed image brightness to make it more readable. SEM images were taken with different modes or mortars or for nanocellulose powders, this is why samples A and B were imaged in high vacuum – such images are not to be compared with the following 3.

  1.  These areas must be addressed by the authors to provide an adequate description of their work

Round 2

Reviewer 1 Report

I think the current revised version seems OK for me

Author Response

Dear reviewer thank you for your suggestion and for you final decision

Best regards

Reviewer 2 Report

The authors have modified the ms in light of the reviewers comments and is now suitable for publication

Author Response

Dear reviewer thank you for your suggestion and for your final decision. Best regards